# Advanced Electrode Coatings Based on Poly-N-Phenylanthranilic Acid Composites with Reduced Graphene Oxide for Supercapacitors

**DOI:** 10.3390/polym15081896

**Published:** 2023-04-15

**Authors:** Sveta Zhiraslanovna Ozkan, Lyudmila Ivanovna Tkachenko, Oleg Nikolaevich Efimov, Galina Petrovna Karpacheva, Galina Vasilevna Nikolaeva, Aleksandr Ivanovich Kostev, Nadejda Nikolaevna Dremova, Evgeny Nikolaevich Kabachkov

**Affiliations:** 1A.V. Topchiev Institute of Petrochemical Synthesis, Russian Academy of Sciences, 29 Leninsky Prospect, Moscow 119991, Russia; 2Federal Research Center of Problems of Chemical Physics and Medicinal Chemistry, Russian Academy of Sciences, 1 Academician Semenov Prospect, Moscow 142432, Russia

**Keywords:** poly-N-phenylanthranilic acid, reduced graphene oxide, hybrid composites, electrode coatings, organic electrolyte

## Abstract

The electrochemical behavior of new electrode materials based on poly-N-phenylanthranilic acid (P-N-PAA) composites with reduced graphene oxide (RGO) was studied for the first time. Two methods of obtaining RGO/P-N-PAA composites were suggested. Hybrid materials were synthesized via in situ oxidative polymerization of N-phenylanthranilic acid (N-PAA) in the presence of graphene oxide (GO) (RGO/P-N-PAA-1), as well as from a P-N-PAA solution in DMF containing GO (RGO/P-N-PAA-2). GO post-reduction in the RGO/P-N-PAA composites was carried out under IR heating. Hybrid electrodes are electroactive layers of RGO/P-N-PAA composites stable suspensions in formic acid (FA) deposited on the glassy carbon (GC) and anodized graphite foil (AGF) surfaces. The roughened surface of the AGF flexible strips provides good adhesion of the electroactive coatings. Specific electrochemical capacitances of AGF-based electrodes depend on the method for the production of electroactive coatings and reach 268, 184, 111 F∙g^−1^ (RGO/P-N-PAA-1) and 407, 321, 255 F∙g^−1^ (RGO/P-N-PAA-2.1) at 0.5, 1.5, 3.0 mA·cm^−2^ in an aprotic electrolyte. Specific weight capacitance values of IR-heated composite coatings decrease as compared to capacitance values of primer coatings and amount to 216, 145, 78 F∙g^−1^ (RGO/P-N-PAA-1_IR_) and 377, 291, 200 F∙g^−1^ (RGO/P-N-PAA-2.1_IR_). With a decrease in the weight of the applied coating, the specific electrochemical capacitance of the electrodes increases to 752, 524, 329 F∙g^−1^ (AGF/RGO/P-N-PAA-2.1) and 691, 455, 255 F∙g^−1^ (AGF/RGO/P-N-PAA-1_IR_).

## 1. Introduction

The use of local renewable energy sources operating in an intermittent mode (solar panels, wind farms) has contributed significantly to the development of energy storage electrochemical systems, such as batteries and supercapacitors (SCs) [1,2,3,4,5,6]. Unlike batteries, where electrode reactions occur within the volume of electrodes and depend on the diffusion rate of electrolyte ions, SCs store energy at the electrode/electrolyte interface due to the formation of an electric double layer (EDL) [7,8,9,10]. A considerable contribution to the energy storage in SCs is also provided by pseudocapacitance resulting from fast redox Faraday reactions [11,12,13,14]. The electrochemical process takes place on the surface, which leads to a higher flow rate due to the absence of diffusion hindrance. The value of pseudocapacitance can exceed significantly the capacitance of the EDL. However, during cycling, pseudocapacitance decreases much faster than EDL capacitance.

SCs are characterized by ultra-high power density and extra-long lifetime, as there are no phase transitions in the electrode materials [15,16,17]. At the same time, they are inferior to batteries in terms of capacitance and energy density [18]. The combination of advantages of batteries and SCs can be achievable in hybrid capacitors. This type of system with two energy storage mechanisms requires the development of highly efficient electrode materials that determine the capacitance, energy density, and specific power of SCs [19,20].

Graphene oxide (GO) and reduced graphene oxide (RGO) are the most common carbon-based materials for SCs [21,22,23]. They easily form composites with redox-active polymers to increase the pseudocapacitance that can be used for energy storage [24]. GO and RGO consist of graphene nanosheets forming nanostacks and have a large specific surface area [25,26,27]. The oxygen-containing functional groups provide good wettability of GO nanostacks surface and are convenient for functionalization [28] and the formation of composites, but lead to high electrical resistance. Depending on the oxidation depth and the moisture content, interplanar distances in GO nanostacks range from 6 to 12 Å, which exceeds significantly the interplanar distance in graphite (0.335 nm). The weakening of van der Waals forces between nanostacks makes it possible to obtain stable suspensions of GO in water and in polar organic solvents.

Along with customary methods of obtaining GO by graphite oxidation in a mixture of acids, the method of electrochemical exfoliation of graphite via the anodic or cathodic treatment of graphite electrodes has also become widespread [29,30]. Anodic treatment proceeds more efficiently and is accompanied by the delamination of particles with a high content of sp^2^-hybridized carbon structures.

Nowadays, electrically conductive polymers (polyaniline (PANI), polypyrrole, and polythiophene) are being used as the redox active component of composite cathode materials in SCs [31,32,33,34,35]. In comparison with inorganic redox-active components, they have low density and a high rate of redox processes. To obtain SC electrode materials, it is necessary to ensure that the pseudocapacitance of the redox active component is combined with the double-layer charging in the supercapacitor at the nanolevel while maintaining the electrical conductivity of the composite material and its highly porous structure. Composite electrode materials are mainly obtained via in situ aniline and other aromatic amine oxidative polymerization in the presence of GO [36,37]. The specific capacitance of PANI nanocomposites with GO or RGO ranges from 210 to 1130 F∙g^−1^.

The process of GO reduction is important because the properties of RGO are similar to those of graphene characterized by a large specific surface area [26,27,38]. Few-layer graphene is characterized by a low oxidation state (~3% O) and a near-zero intensity ratio I_D_/I_G_ = 0.07 [39]. Hydrazine and its derivatives do not react with water, which makes them very attractive for the reduction of aqueous solutions of GO. However, during chemical reduction with hydrazine, graphene nanostacks may aggregate [40], which reduces the porosity and wettability of the material by the electrolyte. The advantages of the electrochemical reduction of GO include being environmentally friendly, highly efficient, energy-saving, and controllable [41]. Electrochemically-reduced GO and its composites are plated on electrodes and used as an excellent electrode material [42,43,44,45,46,47]. A well-known limitation of electrochemical methods of GO reduction is the small size of the electrodes, which makes it difficult to use them for the practical production of electrochemically reduced graphene oxide (ERGO).

The combination of a porous carbon substrate with a high surface area and new electroactive composite coatings in electrode materials makes it possible to balance contributions of double-layer charging and Faraday pseudocapacitance in order to obtain both high capacitance and high charge–discharge currents. Hybrid composites based on conjugated polymers and reduced graphene oxide have been studied exclusively as cathode materials for supercapacitors in aqueous sulfuric acid and alkaline electrolytes. We have not been able to find any references to the study of such composites in organic electrolytes with lithium salts. Such cathode materials in organic electrolytes are the most promising for the creation of hybrid supercapacitors due to the possibility of increasing the supercapacitor voltage and achieving high values of energy density and charge–discharge current.

The present work is the first study of the electrochemical behavior of a cathode material based on a conductive polymer and RGO in a lithium organic electrolyte. In this research paper, the RGO/P-N-PAA composites for electroactive coatings were prepared in two different ways for the first time via in situ oxidative polymerization of monomer (N-PAA) in the presence of GO and from a solution of polymer (P-N-PAA) in DMF containing GO. For GO post-reduction IR heating of synthesized RGO/P-N-PAA composites was carried out. The electrochemical properties of the electroactive hybrid coatings on both a glassy carbon (GC) surface and an AGF substrate in an organic electrolyte (1 M LiClO_4_ in propylene carbonate) were investigated. 

## 2. Experimental

### 2.1. Materials

N-Phenylanthranilic acid (diphenylamine-2-carboxylic acid) (C_13_H_11_O_2_N) (analytical grade), (NH_4_)_2_SO_4_ (Fisher Chemical), sulfuric acid (reagent grade), hydrochloric acid (reagent grade), nitric acid (reagent grade), formic acid (FA) (analytical grade), aqueous ammonia (reagent grade), chloroform (reagent grade), KMnO_4_ (reagent grade), H_2_O_2_ (reagent grade) were used as received without any additional purification. Ammonium persulfate (analytical grade) was purified by recrystallization from distilled water. LiClO_4_ (Aldrich) was dried in a vacuum at 120 °C for 3 days. Propylene carbonate was dried over molecular sieves. The electrolyte from 1 M LiClO_4_ solution in propylene carbonate was stored under argon.

Cold rolling of thermally expanded graphite [48,49] was used to obtain the graphite foil (GF) (Unichimtek, MSU, Russia). The GF strips (0.8 mm thick and 5 × 0.5 cm in size) were anodized in 0.1 M (NH_4_)_2_SO_4_ electrolyte for 4 min at *V* = 3.0 V and *I* = 0.3 A [50]. The GC-2000 glassy carbon plates (NIIgrafit, Moscow, Russia) sized 0.5 × 3 cm was polished with a diamond paste of the ASM-3/2 type. 

The micromechanical exfoliation of graphite oxide was used to obtain a suspension of graphene oxide [51]. A suspension of graphite oxide in water (15 mg/mL) was processed on an ultrasonic disperser for 30 min at 50 °C.

### 2.2. Synthesis of RGO/P-N-PAA

The RGO/P-N-PAA composites were prepared in two different ways.

RGO/P-N-PAA-1 was obtained via in situ oxidative polymerization of N-phenylanthranilic acid (N-PAA) monomer in the water/chloroform heterophase system in an alkaline medium in the presence of GO according to the method described in [52]. The content of graphene oxide is C_GO_ = 20 wt % relative to the monomer weight, C_(NH4)2S2O8_ = 0.2 mol/L, C_monomer_ = 0.1 mol/L, C_NH4OH_ = 0.5 mol/L. The yield of the RGO/P-N-PAA-1 composite is 0.61 g (78.5%).

To synthesize RGO/P-N-PAA-2 composites in the second way, a solution of P-N-PAA in DMF containing GO was prepared. The content of graphene oxide is C_GO_ = 20 and 50 wt % relative to the polymer weight. The GO/P-N-PAA suspension was stirred in an ultrasonic bath for 0.5 h. The RGO/P-N-PAA-2 composites were prepared by removing the solvent at *T* = 60–85 °C and were marked as RGO/P-N-PAA-2.1 and RGO/P-N-PAA-2.2 at C_GO_ = 20 and 50 wt %, respectively. Depending on the synthesis conditions, the RGO/P-N-PAA-2 yield is 87–90%.

### 2.3. Post-Reduction of Graphene Oxide in the RGO/P-N-PAA Composites

For complete graphene oxide reduction, the prepared RGO/P-N-PAA composites were IR-heated with an automated IR heating unit [53] in an Ar at 350 °C for 10 min. The heating rate was 50 °C min^−1^. Composites were marked as RGO/P-N-PAA-1_IR_, RGO/P-N-PAA-2.1_IR_, and RGO/P-N-PAA-2.2_IR_. Depending on the selected synthesis conditions, the RGO/P-N-PAA_IR_ yield is 71–75%.

### 2.4. Electrode Preparation

The electroactive coatings made of suspensions of RGO/P-N-PAA composites prepared before and after IR heating were applied to the GC and AGF substrates. An HD 3200 ultrasonic homogenizer was used to sonicate the suspensions of composites in FA (1.5 wt %). The coating area was 1 cm^2^. 

### 2.5. Electrochemical Measurements

An IPC-Compact P-8 potentiostat (Elins, Russia) was used to record cyclic voltammograms (CV) and galvanostatic charge–discharge curves in the potential range of −0.5–1.4 V. Electrochemical measurements were made in a sealed three-electrode cell in the argon atmosphere in a 1 M LiClO_4_ solution in propylene carbonate. The Pt plate (1 cm^2^) was used as an auxiliary electrode. The Ag/AgCl was used as a reference electrode.

Coulombic efficiency ŋ, specific weight, and surface capacitances C_w_ and C_s_ were calculated from the charge–discharge curves according to the method described in [54]. 

### 2.6. Materials Characterization

TGA thermograms were taken using a Mettler Toledo TGA/DSC1 thermal analyzer (Columbus, OH, USA) in the range of 30–1000 °C in air and in the argon flow.

FE-SEM images were taken using a Zeiss Supra 25 FE-SEM field emission scanning electron microscope (Carl Zeiss AG, Jena, Germany) equipped with an Oxford Instruments X-ray energy-dispersive INCA Energy to identify the elemental composition of the samples. Image resolution is 1–2 nm.

High-resolution XPS spectra were recorded on the Specs PHOIBOS 150 MCD electronic spectrometer (Berlin, Germany). A magnesium anode X-ray tube (Mg*K*_α_ radiation 1253.6 eV) was used.

An HYPERION-2000 IR microscope (Bruker, Karlsruhe, Germany) coupled with the Bruker IFS 66v FTIR spectrometer (Karlsruhe, Germany) in the range of 600–4000 cm^−1^ (150 scans, ZnSe crystal, resolution of 2 cm^−1^) was used to record attenuated total reflection (ATR) FTIR spectra.

Raman spectra were recorded on a Senterra II Raman spectrometer (Bruker, Karlsruhe, Germany) using a laser with a wavelength of 532 nm and a power of 0.25 mW, spectral resolution of 4 cm^−1^.

Electric characteristics of the neat composites were measured using the Miller FPP-5000 4-Point Probe (Fountain Valley, CA, USA).

XRD patterns were recorded using a Difray-401 X-ray diffractometer (Scientific Instruments Joint Stock Company, Saint-Petersburg, Russia) with Bragg–Brentano focusing on Cr*K*_α_ radiation, *λ* = 0.229 nm.

## 3. Results and Discussion

### 3.1. Synthesis and Characterization of RGO/P-N-PAA Composites for Electrode Coatings

Two methods of obtaining hybrid composites based on P-N-PAA and RGO were proposed. Hybrid materials were synthesized for the first time via in situ oxidative polymerization of N-PAA in the presence of GO in the heterophase system in an alkaline medium (RGO/P-N-PAA-1), as well as from a solution of P-N-PAA in DMF containing GO (RGO/P-N-PAA-2). P-N-PAA polyacid, synthesized by the authors for the first time, is well adsorbed on graphene nanosheets due to π-stacking and hydrogen bonding of carboxylic groups [55,56] to oxygen-containing groups on the graphene oxide. With the further recovery of GO and removal of oxygen, the polymer layers prevent aggregation of graphene nanosheets. Figure 1 displays a synthesis scheme of the RGO/P-N-PAA materials. 

The thermal analysis was carried out to determine the temperature of IR heating from TGA data to prevent polymer chain degradation (Figure 2). In the RGO/P-N-PAA-1, the weight loss at 170–220 °C is caused by the removal of the COOH groups that takes place when the composite prepared via oxidative polymerization is heated [52]. The removal of COOH groups is confirmed by high-temperature IR spectroscopy. Comparative analysis of FTIR spectra of the initial polymer and the polymer heated up to 200 °C in air showed that with increasing temperature the intensity of bands at 1682 and 1231 cm^−1^ characterizing COOH groups decreases. In P-N-PAA, the removal of COOH groups begins at temperatures above 150 °C. IR heating of the obtained materials at 350 °C leads to a significant increase in their thermal properties. The IR-heated composites lose half of the initial weight in an inert atmosphere at temperatures higher than 1000 °C (Table 1). At 1000 °C the residue is 69–74% in the RGO/P-N-PAA_IR_ composites.

As seen in (Figure 3), all the main bands characterizing the chemical structure of P-N-PAA remain in the FTIR spectra of RGO/P-N-PAA. In the (ATR) FTIR spectra of composites, bands at 749 and 824 cm^−1^ (RGO/P-N-PAA-1) and at 749 and 820 cm^−1^ (RGO/P-N-PAA-2.1), corresponding to out-of-plane bending vibrations of the δ_C–H_ bonds, indicate the presence of 1,2-disubstituted and 1,2,4-trisubstituted aromatic rings in the structure of polymer component. Intensive bands at 1595 and 1509 cm^−1^ refer to the stretching vibrations of the ν_C–C_ bonds in aromatic rings. The shift of these absorption bands in the (ATR) FTIR spectra of composites indicates the π-π interaction of polymer components with RGO (stacking effect) [57]. In the P-N-PAA, the absorption bands at 1680 and 1231 cm^−1^ characterize the stretching vibrations of ν_C=O_ in COOH groups. The absence of bands at 1680 and 1217 cm^−1^ (RGO/P-N-PAA-1_IR_) and at 1658 and 1213 cm^−1^ (RGO/P-N-PAA-2_IR_) characterizing carboxyl groups, in the (ATR) FTIR spectra of composites subjected to IR heating, is due to the removal of carboxyl groups.

XRD patterns of RGO/P-N-PAA composites (Figure 4) show no sharp peak at 2θ = 17.08° (Cr*K*_α_ radiation) typical of GO [58]. At the same time, the degree of reduction of GO depends on the method of obtaining materials, which is also confirmed by the values of electrical conductivity (Table 2).

IR heating of RGO/P-N-PAA composites prepared in two ways was carried out for post-reduction of GO. The proposed GO reduction method is original and does not require chemical reducing agents. During IR heating, dehydrogenation of the polymer component occurs, as evidenced by the broadening of the main bands in the FTIR spectra of IR-heated composites. The released hydrogen leads to the reduction of GO.

Table 2 provides I_D_/I_G_ data that allow estimating the ratio of sp^3^ and sp^2^ carbon atoms, as well as I_2D_/I_G_ data, that can be used to evaluate the degree of GO reduction in composites. When GO is reduced, the intensity of 2D (the first overtone of the D signal) should increase. In the Raman spectrum of graphene, this I_2D_/I_G_ intensity ratio should be higher than 1. As can be seen, I_2D_/I_G_ does not reach 1 in any of the materials, but it obviously increases, especially in samples subjected to IR heating.

According to XRD data, during the oxidative polymerization of N-PAA in the presence of 20 wt % GO, its partial reduction occurs with the formation of RGO (Figure 4). A peak typical of RGO appears at 2θ = 38.92°. At the same time, the electrical conductivity of RGO/P-N-PAA-1 is σ = 1.8 × 10^−8^ S/cm, whereas that of the primary polymer is σ = 8.8 × 10^−11^ S/cm. IR heating of the resulting composite at 350 °C leads to further reduction of GO. The electrical conductivity of RGO/P-N-PAA-1_IR_ reaches σ = 2.3 × 10^−1^ S/cm, which is higher than the conductivity of GO (2.1 × 10^−4^ S/cm [40]), and I_2D_/I_G_ increases from 0.44 to 0.56 (Table 2). For GO, I_D_/I_G_ = 0.99, I_2D_/I_G_ = 0.33.

Diffraction patterns of composites obtained from a solution of P-N-PAA in DMF containing 20 and 50 wt % GO also show a broad peak typical of RGO at 2θ = 38.92° (Figure 4). As seen, this peak intensity increases both with the growth of GO concentration in the composite and under IR heating. The electrical conductivity is σ = 6.7 × 10^−3^ S/cm (RGO/P-N-PAA-2.1) and σ = 2.7 × 10^−1^ S/cm (RGO/P-N-PAA-2.2), which indicates a higher degree of GO reduction. The I_2D_/I_G_ ratio reaches 0.48 for RGO/P-N-PAA-2.1. It should be noted that when composites are prepared by mixing the polymer and GO in DMF and subsequent heating at 60–85 °C in air to remove the solvent, the products of partial decomposition of both N-PAA oligomers and DMF may participate in the reduction of GO [59]. The electrical conductivity of composites subjected to IR heating for GO post-reduction reaches σ = 2.6 × 10^−1^ S/cm (RGO/P-N-PAA-2.1_IR_) and σ = 1.1 S/cm (RGO/P-N-PAA-2.2_IR_) (Table 2).

### 3.2. Characterization of Hybrid Electrodes

#### 3.2.1. Morphology of Hybrid Electrodes

Stable suspensions of RGO/P-N-PAA composites in FA form electroactive layers on the AGF surface due to penetration and good adhesion [49]. Figure 5 presents FE-SEM images of RGO/P-N-PAA and AGF/RGO/P-N-PAA.

Figure 5a shows an FE-SEM image of the RGO/P-N-PAA-1 composite. As can be seen, the synthesis of the polymer in the presence of 20 wt % GO occurs mainly on the surface of GO nanostacks with the formation of a polymer coating and partial inclusion of the free polymer in the space between nanostacks. 

When RGO/P-N-PAA-1 composite coating from the suspension in FA is applied to AGF (Figure 5f), the ultrasonic treatment causes polymer-coated GO nanostacks to disintegrate with the formation of a highly developed porous surface. In suspensions of composites in FA, the contracted form of P-N-PAA-1 chains becomes more stretched due to interaction with the solvent, which facilitates the compact packing of polymer chains on the surface of graphene nanosheets with improved intermolecular interaction. GO nanosheets coated with a thin polymer layer are curved, probably as a result of delamination of the GO nanostacks during the synthesis when a monomer is wedged between nanosheets. Oxygen-containing groups both on the surface and on the edges of GO nanosheets enable uniform adsorption of monomer molecules with subsequent formation of a thin polymer coating. After IR heating of RGO/P-N-PAA-1, the morphology of the composite changes. Polymer-coated nanostacks disappear, and dense arrays can be observed (Figure 5b). When applied to AGF, the coating surface becomes low-porous and looks like a collection of flat graphite-like arrays (Figure 5g).

Synthesis of RGO/P-N-PAA-2.1 under sonication from a solution of P-N-PAA in DMF containing 20 wt % GO with the following removal of the solvent leads to the formation of a composite with a layered structure (Figure 5c), typical of GO, with possible insertion (intercalation) of the polymer into the interlayer space of GO. When the composite coating is applied to AGF, the curvature of GO structures coated with the polymer (polymer network) is seen (Figure 5h). After the RGO/P-N-PAA-2.1 composite is subjected to IR radiation, the layered prismatic structure with polymer interlayers is preserved (Figure 5d). When RGO/P-N-PAA-2.1_IR_ is applied to AGF, a closely packed coating is formed, consisting of separate irregularly shaped arrays (Figure 5i). However, compared to the AGF/RGO/P-N-PAA-1_IR_ electrode material (Figure 5g), the surface of AGF/RGO/P-N-PAA-2.1_IR_ remains looser (Figure 5i). A stack of graphene nanosheets separated by polymer layers can be seen in the FE-SEM image in some fragments of AGF/RGO/P-N-PAA-2.1_IR_ electrode material (Figure 5i). This confirms the layered structure that is preserved after IR heating. Increasing the GO content from 20 to 50 wt % does not produce any significant change in the morphology of the RGO/P-N-PAA-2.2 composite (Figure 5e,j).

#### 3.2.2. XPS Studies of Hybrid Electrodes

X-ray photoelectron spectroscopy (XPS) studies were used to monitor changes in the binding energy of C1s and N1s peaks in RGO/P-N-PAA composite coatings on AGF. Table 3 shows the data of the elemental composition of the electrode materials surface obtained from the spectra of the overview scanning. The presence of an S2p peak on the surface of the AGF/RGO/P-N-PAA electrode materials points that the polymer component is doped with HSO_4_^−^ ions during precipitation in a solution of H_2_SO_4_.

According to the XPS elemental data, the oxygen content on the electrode materials surface drops significantly after IR heating. The reduced content of oxygen indicates both the reduction of GO and the removal of COOH groups in the structure of the polymer component, which agrees with the FTIR spectroscopy data.

Energy-dispersive X-ray spectroscopy (EDS) elemental mapping method was applied to characterize the element distribution in the RGO/P-N-PAA composites. Figure 6 demonstrates SEM-EDS mapping images of carbon C, nitrogen N, and oxygen O in RGO/P-N-PAA. As seen, the oxygen content in the RGO/P-N-PAA_IR_ composites decreases due to the GO reduction and decarboxylation of the polymer chain. Depending on the synthesis conditions, the content of oxygen drops from 19.0 to 3.7 at%.

Figure 7 presents C1s XPS spectra of GF/P-N-PAA and AGF/RGO/P-N-PAA electrode materials. The C1s XPS spectrum of GF/P-N-PAA can be fitted for 4 components related to carbon atoms in different groups (Figure 7a). A strong peak at 284.5 eV refers to sp^2^-hybridized carbon atoms in aromatic rings. A peak at 288.8 eV is typical of carboxyl groups in the aromatic ring of a polymer chain [60,61].

The same peaks are present in the C1s XPS spectra of AGF/RGO/P-N-PAA electrode materials as in GF/P-N-PAA (Figure 7b,d). A low-intensity peak appears at ~290.4 eV, which characterizes the π-π interaction of graphene nanosheets with the polymer chain [57]. After the resulting composites are IR-heated, the peak typical of the C–O bond is reduced in the C1s XPS spectra of AGF/RGO/P-N-PAA_IR_ (Figure 7c,e,f). At the same time, the π-π interaction in the composite coatings is more pronounced.

Figure 8 presents N1s XPS spectra of the surface of GF/P-N-PAA and AGF/RGO/P-N-PAA electrode materials. The analysis of the N1s XPS spectrum of GF/P-N-PAA showed the absence of quinoimine (=N–) groups, which matches FTIR spectroscopy data. In the N1s XPS spectrum of P-N-PAA coatings on GF, there is no peak at 398.7 eV associated with the C=N binding energy (Figure 8a). The ground state is the state of (–NH–) benzoid amine groups at 400.2 eV. There is also a low-intensity peak at 401.6 eV (N^+^) associated with the doping of the polymer with HSO_4_^−^ ions during its deposition in a ten times excess of H_2_SO_4_ solution. 

The analysis of the N1s XPS spectrum for the AGF/RGO/P-N-PAA-1 composite showed the presence of two almost equal peaks at 400.0 eV (–NH–) and 401.7 eV (N^+^) (Figure 8b). In the N1s XPS spectrum of the AGF/RGO/P-N-PAA-2.1 electrode material (Figure 8d), the low intensity of the peak at 401.6 eV is due to the dedoping of the polymer component in DMF during RGO/P-N-PAA-2 synthesis.

In the N1s XPS spectra of AGF/RGO/P-N-PAA_IR_, a peak at 398.9 eV corresponds to the C=N binding energy, which is consistent with FTIR spectroscopy data. Regardless of the preparation method, IR heating of the RGO/P-N-PAA composite at 350 °C results in the dehydrogenation of phenylenamine structures with the formation of C=N bonds. This leads to the partial removal of carboxyl groups.

In the N1s XPS spectra of AGF/RGO/P-N-PAA_IR_, a sharp decrease and shift of the peak at 401.6 eV (N^+^) is associated with the decrease in the level of polymer component doping during IR heating of the composite material due to charge transfer from the polymer chain to GO during their interaction. 

### 3.3. Electrochemical Behavior of Hybrid Electrodes

The electrochemical behavior of RGO/P-N-PAA electrode coatings on a smooth GC surface and a roughened AGF substrate in an organic electrolyte was studied.

#### 3.3.1. Electrochemical Behavior of GC/RGO/P-N-PAA Electrodes in 1 M LiClO_4_ in Propylene Carbonate

Figure 9 shows the CV curves of the GC/RGO/P-N-PAA electrodes compared to GC/P-N-PAA. The redox transitions of RGO/P-N-PAA coatings are clearly visible on the smooth GC surface in 1 M LiClO_4_ in propylene carbonate at the potential scan rate of 20 mV·s^−1^. The GC contribution to the electrochemical capacitance of coatings is negligible (C_s_ = 0.63 × 10^−3^ F∙cm^−2^) [54]. 

For GC/P-N-PAA, the presence of four anodic peaks at 0.5, 0.7, 0.89, and 1.12 V (Figure 9a) is associated with two different types of redox transitions [54]. The anodic peaks at 0.5 and 0.7 V are due to the redox transitions of P-N-PAA/P-N-PAA^+●^ radical cations. The peaks at 0.89 and 1.12 V are caused by the redox transitions of P-N-PAA^+●^/P-N-PAA^2+^ dications. The cathodic peaks are registered at 0.8 and 0.2 V for the P-N-PAA^2+^/P-N-PAA^+●^ and P-N-PAA/P-N-PAA^+●^ redox transitions. 

The CV of the GC/RGO/P-N-PAA-1 composite electrode demonstrates indistinct peaks: a broad anodic peak in the region from 0.4 to 1.0 V with the maximum current at 0.67 V, and a cathodic peak in the region from −0.06 to 0.35 V that refer to redox transitions in P-N-PAA (Figure 9b). The capacitance of the RGO/P-N-PAA-1 composite coating is mostly contributed by the capacitance of the electric double layer. In addition, unlike in the case of neat P-N-PAA, for RGO/P-N-PAA-1 composite coating, there is no drop in anodic and cathodic currents during cycling. The active mass barely dissolves in the electrolyte, which is due to both π-stacking of the polymer with graphene planes [57] and good adhesion of the composite coating to the smooth GC surface. 

The CV of the GC/RGO/P-N-PAA-1_IR_ electrode material reflects changes in the composite after IR heating (Figure 9c). The CV of the GC/RGO/P-N-PAA-1_IR_ electrode acquires a quasi-rectangular shape, which is typical only of double-layer capacitance. Few or no redox transitions of the polymer component in the composite are observed.

Figure 9d shows the CV curve of the GC/RGO/P-N-PAA-2.1 electrode material. The capacitance of the coating is determined by a significant contribution of Faraday pseudocapacitance due to redox processes occurring in the polymer. The introduction of GO results in an increase in the polymer electroactive surface at the interface with the electrolyte. The CV of the GC/RGO/P-N-PAA-2.1 electrode material shows a pair of redox peaks with oxidation potentials of 0.96 V and reduction potentials of 0.73 V. In the anode region, currents decrease during cycling, and after 7 cycles the CV of the GC/RGO/P-N-PAA-2.1 electrode is stable. The decrease in currents in the anode region may be due to the partial dissolution of the polymer not bound to RGO. After RGO/P-N-PAA-2.1 is IR processed, the CV shape of the composite coating changes (Figure 9e), reflecting a larger contribution of double-layer capacitance while retaining redox transitions typical of the polymer. The CV of the GC/RGO/P-N-PAA-2.1_IR_ electrode shows that anodic and cathodic currents increase during cycling. The process becomes more reversible, and the difference in the potentials of the cathodic and anodic peaks is 100 mV. 

Figure 10 demonstrates the galvanostatic charge–discharge curves of the GC/RGO/P-N-PAA electrodes at a charge–discharge current density of 0.1 and 0.5 mA∙cm^2^ in 1 M LiClO_4_ in propylene carbonate. Table 4 shows electrochemical capacitances of the GC/RGO/P-N-PAA electrodes, obtained by galvanostatic charge–discharge dependencies.

When comparing the CV and capacitance data obtained from the charge–discharge curves of these composites after IR heating, it turned out that introducing more RGO decreases the capacitance characteristics of the electrode. Therefore, the electrodes with RGO(50%)/P-N-PAA-2.2 coating was not investigated, assuming that the capacitance of the initial coating would be lowered compared to RGO(20%)/P-N-PAA-2.1 because of the reduced Faraday capacitance contribution from the polymer since its content in this composite was reduced by 30%.

The calculated electrochemical capacitances at 0.1 and 0.5 mA∙cm^−2^ for the RGO/P-N-PAA-1 composite coating on GC are 32 and 24 F∙g^−1^, respectively (Figure 10a). After IR heating, electrochemical capacitances of the GC/RGO/P-N-PAA-1_IR_ electrode material amount to 12.0 and 10.2 F∙g^−1^, respectively (Figure 10b). According to FE-SEM data, the roughness of the coating in the RGO/P-N-PAA-1_IR_ is reduced, as wrinkled GO sheets straighten out (Figure 5a,b). During IR heating of the RGO/P-N-PAA-1 composite, the reduction of GO occurs with the removal of oxygen-containing functional groups, which leads to the aggregation of graphene nanosheets and a decrease in the surface area of the electrode/electrolyte interface. Apparently, it is the densification of the RGO/P-N-PAA-1_IR_ coating, despite the increase in its electronic conductivity, that affects the decrease in the capacitance properties of the electrode material.

The charge–discharge dependencies of the RGO/P-N-PAA-2.1 composite coating on GC are represented by non-linear and asymmetric charge–discharge curves (Figure 10c). The capacitance is mainly determined by Faraday reactions of the polymer component in the composite, and it decreases significantly with the increase in the current density. Electrochemical capacitances of the GC/RGO/P-N-PAA-2.1 electrode, obtained by galvanostatic charge–discharge with currents of 0.1 and 0.5 mA∙cm^−2^, amount to 17.5 and 9.5 F∙g^−1^, which is less than those of GC/RGO/P-N-PAA-1 (Table 4). After IR heating of the RGO/P-N-PAA-2.1 composite, electrochemical capacitances of the GC/RGO/P-N-PAA-2.1_IR_ electrode at the same charge–discharge currents are 29.8 and 22.2 F∙g^−1^ (Figure 10d), which is comparable with capacities obtained for the RGO/P-N-PAA-1 composite coating.

Coulombic efficiency ŋ of charge–discharge rises from 80% (for neat GC/RGO/P-N-PAA-2.1) to 98% (for IR-heated GC/RGO/P-N-PAA-2.1_IR_). During IR heating of RGO/P-N-PAA-2.1, post-reduction of GO with the formation of a highly conductive framework occurs, as well as optimal densification of the composite while maintaining its porous structure, which creates conditions for both ion and electron transport. Morphological features of the RGO/P-N-PAA-2.1 composite, namely the layered structure (intercalation of polymer layers between graphene structures) do not allow them to aggregate. This difference in capacitance properties of the electrodes is apparently due to the difference in the packing density of the obtained RGO/P-N-PAA-1_IR_ and RGO/P-N-PAA-2.1_IR_ composite coatings. 

#### 3.3.2. Electrochemical Behavior of AGF/RGO/P-N-PAA Electrodes in 1 M LiClO_4_ in Propylene Carbonate

The redox transitions of RGO/P-N-PAA coatings are not clearly visible on the loose roughened AGF surface at the potential scan rate of 20 mV·s^−1^. Redox transitions become noticeable on CV at 5 mV·s^−1^. Figure 11 and Figure 12 present the CV curves and the galvanostatic charge–discharge dependencies of the AGF/P-N-PAA and AGF/RGO/P-N-PAA electrodes at 0.5, 1.5 и 3.0 mA∙cm^−2^. The calculated data obtained from the galvanostatic charge–discharge dependencies of the AGF-based electrode materials related to the weight of the coatings are given in Table 5. 

When electroactive coatings made from FA suspensions of RGO/P-N-PAA composites prepared before and after IR heating are applied to loosened functionalized AGF strips, a significant part of the suspension is absorbed into the boundary layer. No coloring of the electrolyte points to good adhesion of the coatings to the AGF surface. At the same time, there is a sharp enhancement in the electrochemical properties of the AGF/RGO/P-N-PAA electrodes.

The CV of the AGF/P-N-PAA electrode shows two anodic waves at 0.54 and 1.0 V and their corresponding broad cathodic ones at 0.73 and 0.25 V associated with redox transitions in the polymer. The CV of the AGF/RGO/P-N-PAA electrodes shows no redox transitions due to the preferential contribution of the electric double-layer charging to the specific capacitance (Figure 11). Coulombic charge–discharge efficiency ŋ of AGF/RGO/P-N-PAA electrodes is close to 100%.

At a charge–discharge current of 0.5 mA·cm^−2^, the AGF/P-N-PAA and AGF/RGO/P-N-PAA-1 hybrid electrodes demonstrate stable work after the tenth cycle (Figure 12a,b). For P-N-PAA and RGO/P-N-PAA-1 coatings, losses of the initial surface-specific capacitance after the first 10 cycles are 11 and 4% while Coulombic efficiency ŋ remains 100%. At commensurate weights of polymer (0.49 mg) and composite (0.37 mg) coatings to AGF, specific weight capacitance C_w_ is 202, 106, 63 F∙g^−1^ (AGF/P-N-PAA) [54] and 268, 184, 111 F∙g^−1^ (AGF/RGO/P-N-PAA-1) at charge–discharge currents of 0.5, 1.5 and 3.0 mA·cm^−2^ (Table 5). This increase in specific capacitance with the introduction of GO can be associated both with an increase in electronic conductivity due to partial reduction of GO during polymerization, and ionic conductivity due to an increase in the interfacial surface at the interface with the electrolyte.

For RGO/P-N-PAA-2.1 composite coatings, at a charge–discharge current of 0.5 mA·cm^−2^, on the second day of operation, after 10 cycles, specific capacitance decreased by 7%. Starting from the third day, the AGF/RGO/P-N-PAA-2.1 electrode works stably. At the same time, the electrochemical capacitance of RGO/P-N-PAA-2.1 composite coatings to AGF remains quite high, at 407, 321, and 255 F∙g^−1^ at 0.5, 1.5, and 3.0 mA·cm^−2^, respectively. With a decrease in the weight of the applied coating to 0.21 mg, the specific electrochemical capacitance of the AGF/RGO/P-N-PAA-2.1 electrode material rises to 752, 524, 329 F∙g^−1^. At high charge–discharge currents of 1.5 and 3.0 mA·cm^−2^, composite RGO/P-N-PAA-2.1 coatings on AGF function stably over 50 cycles.

A sharp (almost 1.5 times) increase in specific capacitance of the RGO/P-N-PAA-2.1 composite coating obtained from a solution of P-N-PAA in DMF containing GO, in comparison with the RGO/P-N-PAA-1 composite obtained via in situ oxidative polymerization, may be caused by increased electrochemical activity due to morphological features (introduction and interaction (π-stacking) of polymer chains with graphene nanosheets), as well as higher conductivity of the composite. According to XRD data (Figure 4, curves 5 and 6), during the synthesis of RGO/P-N-PAA-2.1 and RGO/P-N-PAA-2.2 composites by mixing the polymer and GO in DMF and subsequent heating at 60–85 °C to remove the solvent, GO is reduced to a greater extent with the formation of a conducting framework in comparison with the RGO/P-N-PAA-1 composite obtained via in situ polymerization. 

IR-heated composite coatings to AGF were investigated at commensurate weights. The electrodes demonstrate stable operation over 50 cycles at charge–discharge currents of 0.5, 1.5, and 3.0 mA·cm^−2^ (Figure 12). However, the specific weight capacitance of electrodes falls down to 216, 145, 78 F∙g^−1^ (AGF/RGO/P-N-PAA-1_IR_), 377, 291, 200 F∙g^−1^ (AGF/RGO/P-N-PAA-2.1_IR_) and 324, 218, 142 F∙g^−1^ (AGF/RGO/P-N-PAA-2.2_IR_) (Table 5). This difference in the capacitance properties of composite coatings obtained before and after IR heating is apparently explained by the difference in the density of the obtained materials.

The capacitance is increased when the coatings are applied with a low weight. Comparison of both AGF/RGO/P-N-PAA-1, AGF/RGO/P-N-PAA-2.1 and AGF/RGO/P-N-PAA-1IR, AGF/RGO/P-N-PAA-2.1IR electrodes with comparable weights of 0.37, 0.29, 0.38, and 0.35 mg coatings supports our conclusions about higher capacitance characteristics for composite coating prepared from a solution of P-N-PAA polymer in DMF containing GO.

Figure 13 presents charge–discharge dependencies of four AGF/RGO/P-N-PAA-1_IR_ electrodes with different weights (0.11–0.62 mg) of composite coatings applied to the AGF surface.

The analysis of AGF/RGO/P-N-PAA-1_IR_ charge–discharge curves implies that the discharge times of the electrodes, regardless of the weight of applied coatings, are close. This may indicate that the major contribution to the electrode electrochemical capacitance is provided by a thin coating layer bonded directly to the AGF surface. When a thin layer of RGO/P-N-PAA-1_IR_ composite coating is applied, the suspension is absorbed (drawn in) into the AGF near-surface layer and coats pore walls, leaving the pores open, so they are filled with the electrolyte. In the thin electroactive coating, polymer chains on the surface of graphene nanosheets undergo a fast electrochemical doping/dedoping process due to unhindered electron transfer and ion migration. With an increase in the weight of the applied RGO/P-N-PAA-1_IR_ composite coating from 0.11 to 0.62 mg, specific capacitances drop from 691 to 123 F∙g^−1^ at the discharge current of 0.5 mA·cm^−2^ (Table 5) due to diffusion limitations. Dispersed particles that block pores, preventing the diffusion of ions from the electrolyte, are aggregated.

## 4. Conclusions

Two methods of obtaining composites for electrode coatings were proposed via in situ oxidative polymerization of N-PAA monomer in the presence of GO (RGO/P-N-PAA-1) and from a solution of P-N-PAA polymer in DMF containing GO (RGO/P-N-PAA-2). Composite materials were IR-heated for GO post-reduction. The electrochemical behaviors of the advanced electrodes based on the RGO/P-N-PAA composite coatings on smooth (GC) and roughened (AGF) substrates in 1 M LiClO_4_ electrolyte in propylene carbonate were studied. Using the AGF as a current collector leads to a significant improvement in the electrochemical capacitance and stability of the coatings due to the penetration of RGO/P-N-PAA suspensions in FA into modified substrate pores. Depending on the preparation method and the weight of electroactive coatings, specific electrochemical capacitances of hybrid electrodes reach 752, 524, 329 F∙g^−1^ for AGF/RGO/P-N-PAA-2.1 (0.21 mg) and 691, 455, 255 F∙g^−1^ for AGF/RGO/P-N-PAA-1_IR_ (0.11 mg) at charge–discharge currents of 0.5, 1.5, 3.0 mA·cm^−2^. The major input to the electrochemical capacitance is provided by the electric double-layer charging. The high electrochemical capacitance of the AGF/RGO/P-N-PAA electrodes in an organic electrolyte makes them promising as a cathode material for SCs with increased voltage.

## Figures and Tables

**Figure 1 polymers-15-01896-f001:**
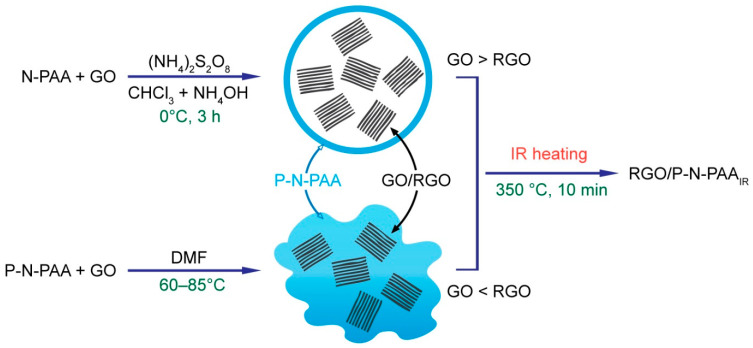
Scheme of the RGO/P-N-PAA materials synthesis.

**Figure 2 polymers-15-01896-f002:**
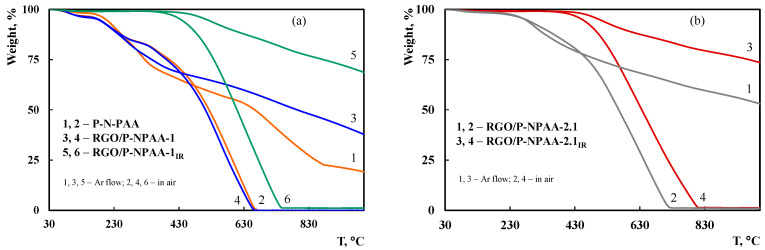
TGA thermograms of RGO/P-N-PAA, prepared via oxidative polymerization (**a**) and from a solution (**b**), at heating up to 1000 °C in air (2, 4, 6) and in the argon flow (1, 3, 5).

**Figure 3 polymers-15-01896-f003:**
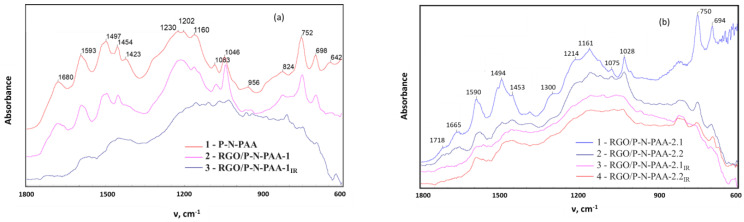
(ATR) FTIR spectra of P-N-PAA (1a) and RGO/P-N-PAA, prepared via oxidative polymerization (**a**) and from a solution (**b**).

**Figure 4 polymers-15-01896-f004:**
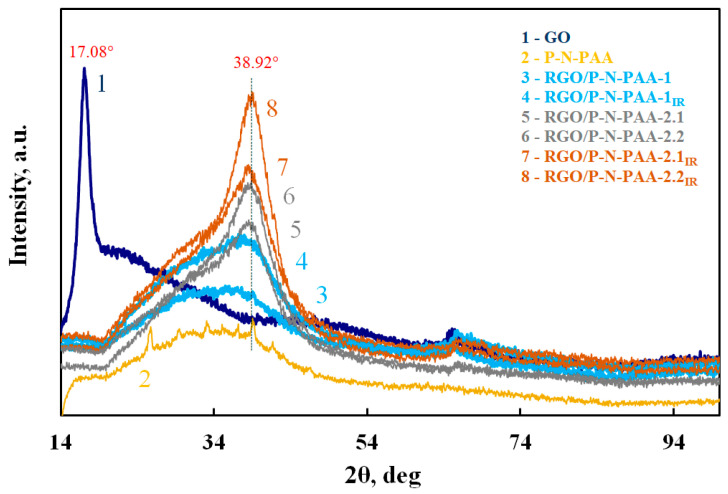
XRD patterns of GO (1), P-N-PAA (2), and RGO/P-N-PAA (3–8).

**Figure 5 polymers-15-01896-f005:**
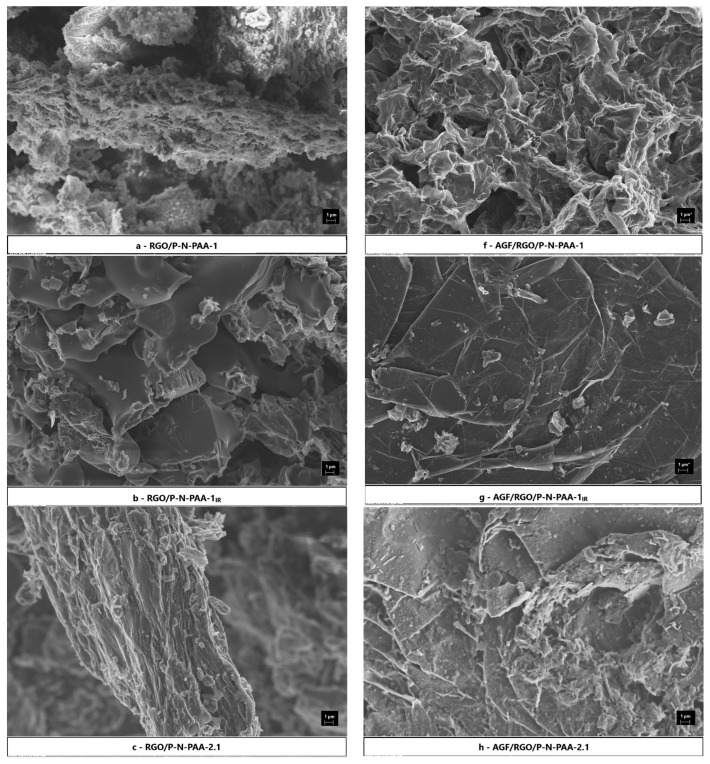
FE-SEM images of RGO/P-N-PAA (**a**–**e**) and AGF/RGO/P-N-PAA (**f**–**j**).

**Figure 6 polymers-15-01896-f006:**
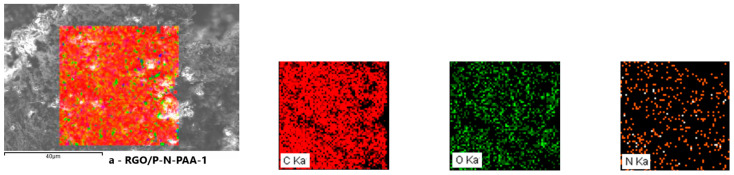
SEM-EDS mapping images of carbon C, nitrogen N, and oxygen O in RGO/P-N-PAA.

**Figure 7 polymers-15-01896-f007:**
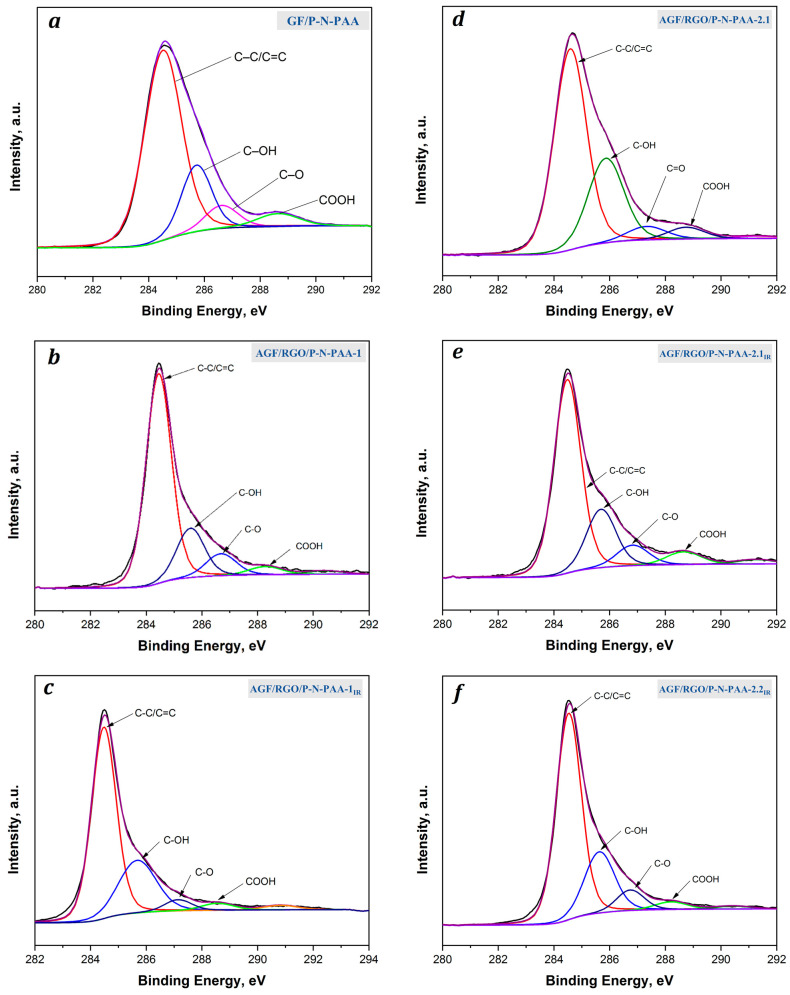
C1s XPS spectra of GF/P-N-PAA-1 (**a**) and AGF/RGO/P-N-PAA before (**b**,**d**) and after IR heating (**c**,**e**,**f**).

**Figure 8 polymers-15-01896-f008:**
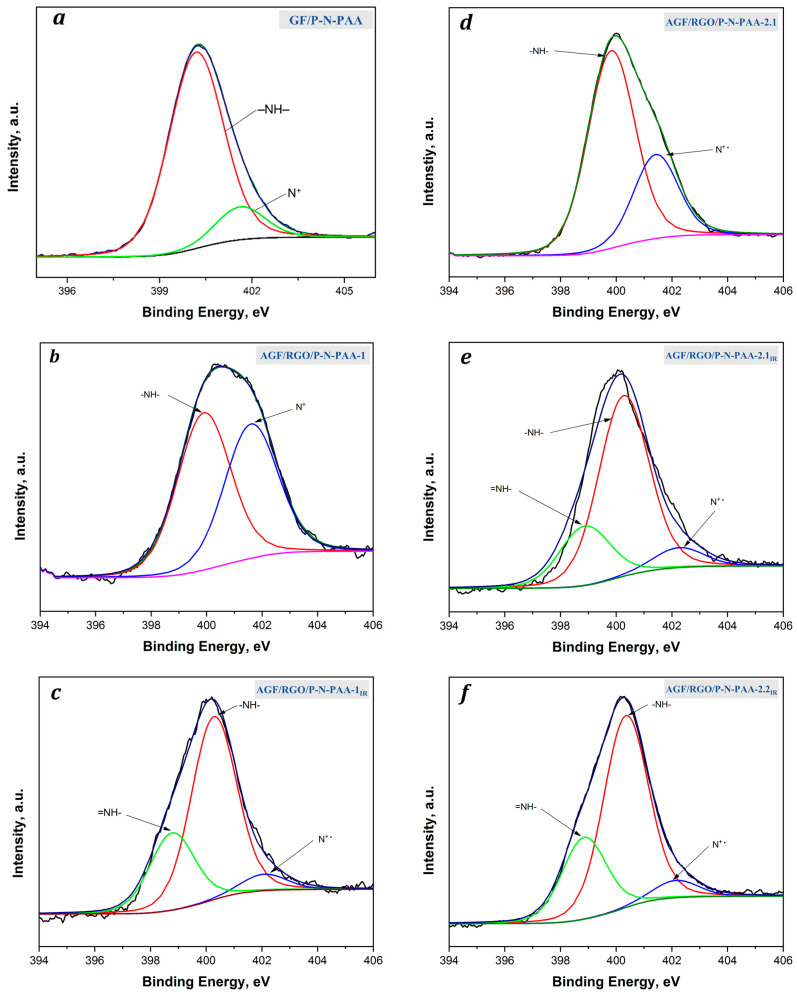
N1s XPS spectra of GF/P-N-PAA-1 (**a**) and AGF/RGO/P-N-PAA before (**b**,**d**) and after IR heating (**c**,**e**,**f**).

**Figure 9 polymers-15-01896-f009:**
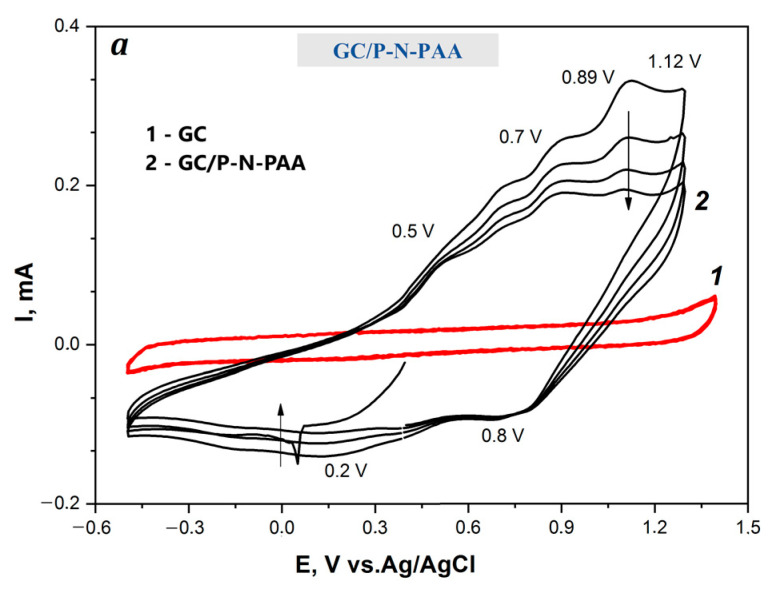
(**a**) CV curves of GC (1a), GC/P-N-PAA (2a), and GC/RGO/P-N-PAA (**b**–**e**) at the potential scan rate of 20 mV·s^−1^.

**Figure 10 polymers-15-01896-f010:**
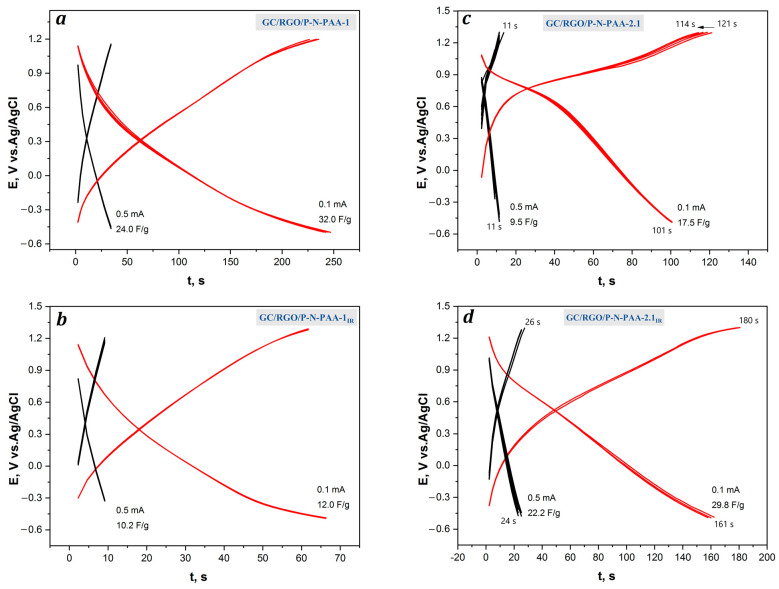
Galvanostatic charge–discharge dependencies of the GC/RGO/P-N-PAA electrodes (**a**–**d**) at 0.1 and 0.5 mA∙cm^−2^.

**Figure 11 polymers-15-01896-f011:**
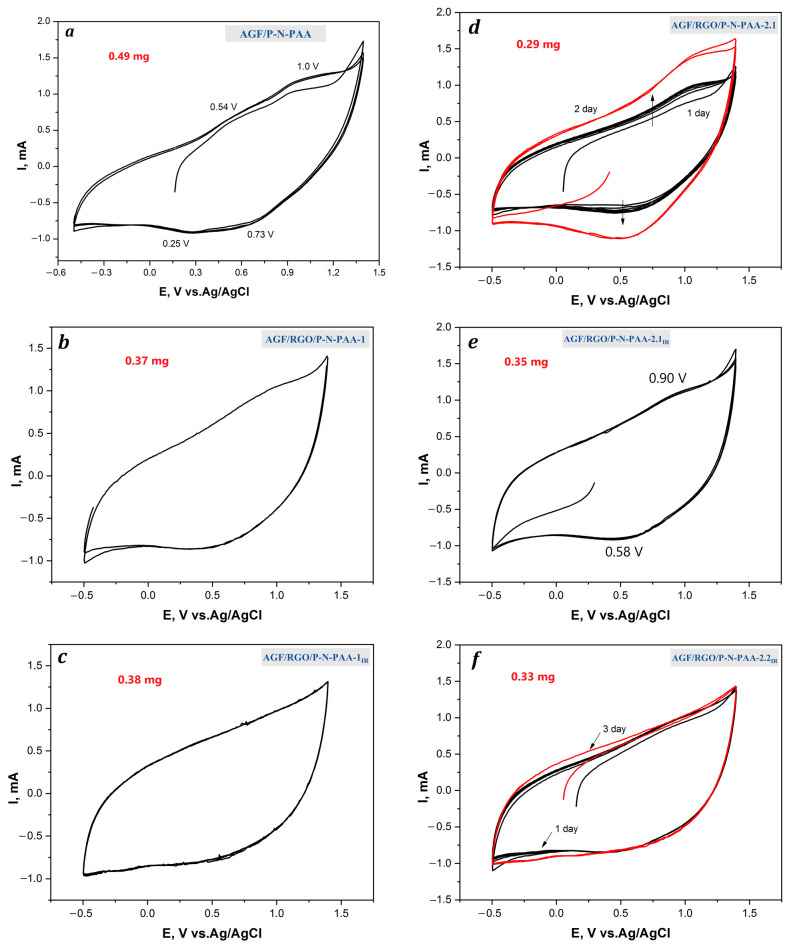
CV curves of the AGF/P-N-PAA (**a**) and AGF/RGO/P-N-PAA before (**b**,**d**) and after IR heating (**c**,**e**,**f**) at 5 mV·s^−1^.

**Figure 12 polymers-15-01896-f012:**
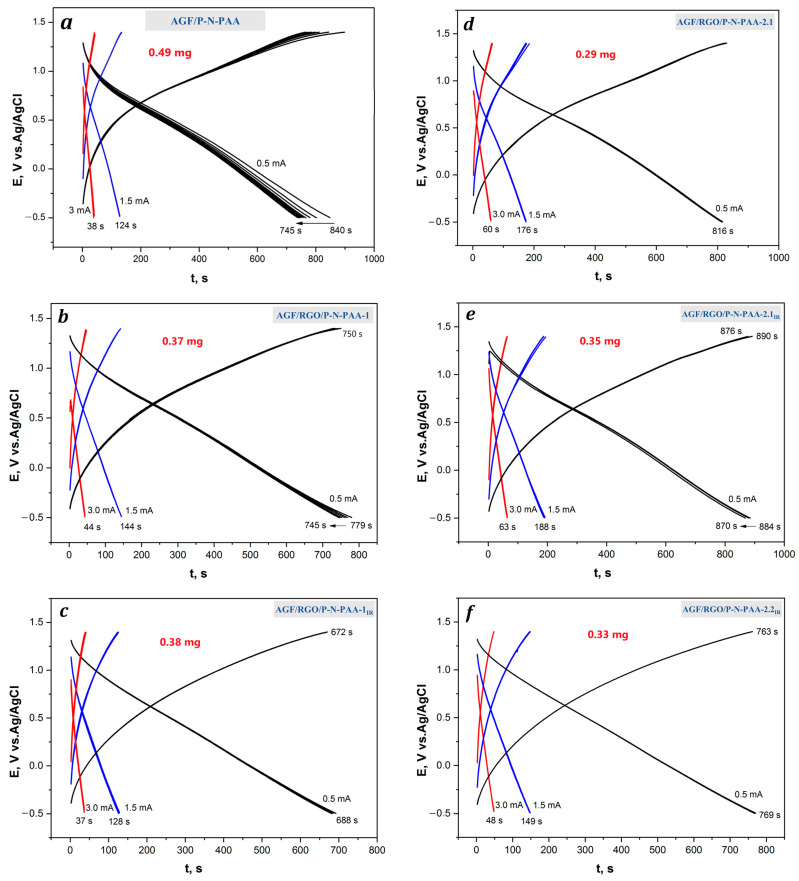
Galvanostatic charge–discharge dependencies of the AGF/P-N-PAA (**a**) and AGF/RGO/P-N-PAA before (**b**,**d**) and after IR heating (**c**,**e**,**f**) at 0.5, 1.5 и 3.0 mA∙cm^−2^.

**Figure 13 polymers-15-01896-f013:**
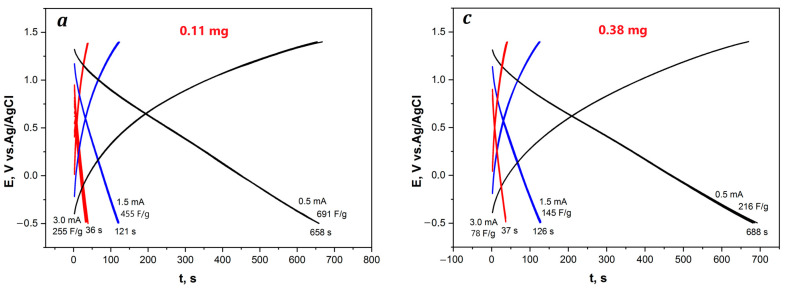
Galvanostatic charge–discharge dependencies of the AGF/RGO/P-N-PAA-1_IR_ electrodes with coatings weight of 0.11 (**a**), 0.21 (**b**), 0.38 (**c**) and 0.62 mg (**d**).

**Table 1 polymers-15-01896-t001:** Thermal properties of nanocomposites.

Materials	Property			
	* *T*_5%_, °C	** *T*_50%_, °C	Weight Loss at 350 °C, %	*** Residue, %
PDPAC	185/205	523/663	20/29	20
RGO/P-N-PAA-1	183/180	511/794	19/26	38
RGO/P-N-PAA-1_IR_	436/515	609/>1000	1/1	69
RGO/P-N-PAA-2.1	279/276	565/>1000	11/14	53
RGO/P-N-PAA-2.1_IR_	453/505	636/>1000	1/1	74

* *T*_5%_, ** *T*_50%_—5 and 50% weight losses (Air/Ar), *** residue at 1000 °C (Ar).

**Table 2 polymers-15-01896-t002:** The conductivity values of materials.

Materials	I_D_/I_G_	I_2D_/I_G_	σ, S/cm
P-N-PAA	0.93	0.43	8.8 × 10^−11^
RGO/P-N-PAA-1	0.92	0.44	1.8 × 10^−8^
RGO/P-N-PAA-1_IR_	0.89	0.56	2.3 × 10^−1^
RGO/P-N-PAA-2.1	0.95	0.48	6.7 × 10^−3^
RGO/P-N-PAA-2.2	0.91	0.50	2.7 × 10^−1^
RGO/P-N-PAA-2.1_IR_	0.85	0.51	2.6 × 10^−1^
RGO/P-N-PAA-2.2_IR_	0.82	0.57	1.1

**Table 3 polymers-15-01896-t003:** XPS elemental composition of hybrid electrodes.

Hybrid Electrodes	C1s, at%	O1s, at%	N1s, at%	S2p, at%
GF/P-N-PAA	81.79	12.19	4.97	1.05
AGF/RGO/P-N-PAA-1	79.02	13.14	5.12	2.72
AGF/RGO/P-N-PAA-1_IR_	85.23	8.35	4.12	1.42
AGF/RGO/P-N-PAA-2.1	73.53	15.69	6.40	4.39
AGF/RGO/P-N-PAA-2.1_IR_	79.73	11.64	4.14	4.49
AGF/RGO/P-N-PAA-2.2_IR_	86.52	7.59	4.24	1.44

**Table 4 polymers-15-01896-t004:** Electrochemical characteristics of GC/RGO/P-N-PAA electrodes.

Hybrid Electrodes	Coatings Weight, mg	Discharge Current Density I_charge–discharge_, mA∙cm^−2^	Coating Specific Weight Capacitance C_w_, F∙g^−1^
GC/RGO/P-N-PAA-1	0.45	0.1 0.5	32.0 24.0
GC/RGO/P-N-PAA-1_IR_	0.30	0.1 0.5	12.0 10.2
GC/RGO/P-N-PAA-2.1	0.32	0.1 0.5	17.5 9.5
GC/RGO/P-N-PAA-2.1_IR_	0.30	0.1 0.5	29.8 22.2

**Table 5 polymers-15-01896-t005:** Electrochemical characteristics of AGF/RGO/P-N-PAA electrodes in 1 M LiClO_4_ in propylene carbonate.

Hybrid Electrodes	Coatings Weight, mg	Discharge Current Density I_charge–discharge_, mA∙cm^−2^	Electrode Specific Surface Capacitance C_s_, F∙cm^−2^	Coating Specific Weight Capacitance C_w_, F∙g^−1^
AGF/P-N-PAA	0.49	0.5 1.5 3.0	0.196 0.098 0.031	202 106 63
AGF/RGO/P-N-PAA-1	0.37	0.5 1.5 3.0	0.196 0.114 0.070	268 184 111
AGF/RGO/P-N-PAA-1_IR_	0.11	0.5 1.5 3.0	0.173 0.096 0.057	691 455 255
	0.21	0.5 1.5 3.0	0.177 0.105 0.062	381 281 157
	0.38	0.5 1.5 3.0	0.182 0.101 0.058	216 145 78
	0.62	0.5 1.5	0.173 0.092	123 74
AGF/RGO/P-N-PAA-2.1	0.21	0.5 1.5 3.0	0.255 0.156 0.098	752 524 329
	0.29	0.5 1.5 3.0	0.215 0.139 0.095	407 321 255
AGF/RGO/P-N-PAA-2.1_IR_	0.35	0.5 1.5 3.0	0.229 0.148 0.099	377 291 200
AGF/RGO/P-N-PAA-2.2_IR_	0.33	0.5 1.5 3.0	0.200 0.118 0.076	324 218 142

## Data Availability

Not applicable.

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
