# Peer review of "Advanced Electrode Coatings Based on Poly-N-Phenylanthranilic Acid Composites with Reduced Graphene Oxide for Supercapacitors"

_polymers, 2023, doi:10.3390/polym15081896_

Round 1

Reviewer 1 Report

I find the manuscript very well written, the results are quite interesting and are worthy to be published.

As authors recognized there are some modifications of morphology fo the composites after IR heating, also the chemical signature change due the same( less of N and COOH groups). The temperature reached via IR heating , i.e. 350°C seems to be high for the polymers.

Is it possible for authors to make thermogravimetric analysis in order get more information about the (GO-polymer) and (rGO-polymer) ratio and changing of polymer structure?

-Please ensure that all abbreviations used in the text are described when they are used for the first time.

Author Response

The authors are grateful to the reviewer for constructive and valuable comments on the manuscript. Please find below our answers to the comments.

Comments and Suggestions for Authors

- As authors recognized there are some modifications of morphology of the composites after IR heating, also the chemical signature change due the same( less of N and COOH groups). The temperature reached via IR heating , i.e. 350°C seems to be high for the polymers.

Is it possible for authors to make thermogravimetric analysis in order get more information about the (GO-polymer) and (rGO-polymer) ratio and changing of polymer structure?

The thermal analysis was carried out to determine the temperature of IR heating from TGA data to prevent polymer chain degradation (Figure 2). In the RGO/P-N-PAA-1, the weight loss at 170–220 °С is associated with the removal of the COOH groups that takes place when the composite prepared via oxidative polymerization is heated [52]. The removal of COOH groups is confirmed by high-temperature IR spectroscopy. Comparative analysis of FTIR spectra of the initial polymer and the polymer heated up to 200 °C in air showed that with increasing temperature the intensity of bands at 1682 and 1231 cm-1 characterizing COOH groups decreases. In P-N-PAA, the removal of COOH groups begins at temperatures above 150 °C.

IR heating of the obtained materials at 350 °C leads to a significant increase in their thermal properties. The IR-heated composites lose half of the initial weight in an inert atmosphere at temperatures above 1000 °C (Table 1). At 1000 °C the residue is 69–74% in the RGO/P-N-PAAIR composites.

During IR heating, dehydrogenation of the polymer component occurs, as evidenced by the broadening of the main bands in the FTIR spectra of IR heated composites. The released hydrogen leads to the reduction of GO. According to the XPS elemental data, the oxygen content on the electrode materials surface drops significantly after IR heating. The reduced content of oxygen indicates both the reduction of GO and the removal of COOH groups in the structure of the polymer component. In the N1s XPS spectra of AGF/RGO/P-N-PAAIR, a peak at 398.9 eV corresponds to the C=N binding energy.

Relevant corrections are included in the Thermal Properties section in colored characters.  

- Please ensure that all abbreviations used in the text are described when they are used for the first time.

Proofreading of the manuscript was carried out.

Reviewer 2 Report

This paper describes the application of a composite of graphene oxide (GO) and a polyaniline derivative (P-N-PAA) to supercapacitors.

This paper employs a unique method of synthesizing reduced graphene oxide (rGO) by reducing GO with IR heating.

However, the IR heating might affect not only GO but also polymers.

Perform thermal analysis of P-N-PAA alone and prove that P-N-PAA does not decompose even with IR heating. Also, prove from the NMR spectrum that P-N-PAA maintains its original chemical structure after heating. In addition, prove through CV that the P-N-PAA alone after heating exhibits the same electrochemical properties as the sample before heating.

Also, while this paper features the use of IR heating as a clean reduction method for graphene oxide, there have been previous reports of clean reduction of GO by electrochemical methods (Carbon 662 (2914); RSC Adv. 86855 (2015); Electrochem. Commun. 65 (2017)). The authors must be sure to cite all of these references and clearly distinguish the differences from the present paper.

The composition of the complex is analyzed by SEM, but it is difficult to determine by appearance alone; perform EDX measurements to show where the rGO and P-N-PAA are distributed.

Table 2, Figure 5, Figure 8, Figure 9, and Table 4 do not show data for AGF/RGO/P-N-PAA-2.2 before IR heating. Include these data for comparison purposes.

Many studies have reported the application of such composites to capacitors. Discuss the numerical comparison of the superiority of the capacitors prepared by this method compared to previous reports.

Unless these are corrected, this paper should not be accepted for publication in this journal.

Author Response

- Paper employs a unique method of synthesizing reduced graphene oxide (rGO) by reducing GO with IR heating. However, the IR heating might affect not only GO but also polymers.

Perform thermal analysis of P-N-PAA alone and prove that P-N-PAA does not decompose even with IR heating. Also, prove from the NMR spectrum that P-N-PAA maintains its original chemical structure after heating. In addition, prove through CV that the P-N-PAA alone after heating exhibits the same electrochemical properties as the sample before heating.

The thermal analysis was carried out to determine the temperature of IR heating from TGA data to prevent polymer chain degradation (Figure 2). In the RGO/P-N-PAA-1, the weight loss at 170–220 °С is associated with the removal of the COOH groups that takes place when the composite prepared via oxidative polymerization is heated [52]. The removal of COOH groups is confirmed by high-temperature IR spectroscopy. Comparative analysis of FTIR spectra of the initial polymer and the polymer heated up to 200 °C in air showed that with increasing temperature the intensity of bands at 1682 and 1231 cm-1 characterizing COOH groups decreases. In P-N-PAA, the removal of COOH groups begins at temperatures above 150 °C.

IR heating of the obtained materials at 350 °C leads to a significant increase in their thermal properties. The IR-heated composites lose half of the initial weight in an inert atmosphere at temperatures above 1000 °C (Table 1). At 1000 °C the residue is 69–74% in the RGO/P-N-PAAIR composites.

During IR heating, dehydrogenation of the polymer component occurs, as evidenced by the broadening of the main bands in the FTIR spectra of IR heated composites. The released hydrogen leads to the reduction of GO. According to the XPS elemental data, the oxygen content on the electrode materials surface drops significantly after IR heating. The reduced content of oxygen indicates both the reduction of GO and the removal of COOH groups in the structure of the polymer component. In the N1s XPS spectra of AGF/RGO/P-N-PAAIR, a peak at 398.9 eV corresponds to the C=N binding energy.

Relevant corrections are included in the Thermal Properties section in colored characters.

- Also, while this paper features the use of IR heating as a clean reduction method for graphene oxide, there have been previous reports of clean reduction of GO by electrochemical methods (Carbon 662 (2914); RSC Adv. 86855 (2015); Electrochem. Commun. 65 (2017)). The authors must be sure to cite all of these references and clearly distinguish the differences from the present paper. 

A well-known limitation of electrochemical methods of GO reduction is the small size of the electrodes, which makes it difficult to use them for the practical production of electrochemically reduced graphene oxide.

Proposed references were cited. Appropriate additions were introduced into the text in colored characters.

- The composition of the complex is analyzed by SEM, but it is difficult to determine by appearance alone; perform EDX measurements to show where the rGO and P-N-PAA are distributed. 

EDX measurements were performed and introduced into the text in colored characters.

- Table 2, Figure 5, Figure 8, Figure 9, and Table 4 do not show data for AGF/RGO/P-N-PAA-2.2 before IR heating. Include these data for comparison purposes.

The RGO(20%)/P-N-PAA-2.1 and RGO(50%)/P-N-PAA-2.2 composites were obtained by the same method and differ only in RGO content. When comparing the CV and capacitance data obtained from the charge-discharge curves of these composites after IR heating, it turned out that introducing more RGO decreases the capacitance characteristics of the electrode. Therefore, the AGF/RGO(50%)/P-N-PAA-2.2 electrode was not investigated, assuming that the capacitance of the initial coating would be lowered compared to RGO(20%)/P-N-PAA-2.1 because of the reduced Faraday capacitance contribution from the polymer, since its content in this composite was reduced by 30%.

The data for the AGF/RGO(20%)/P-N-PAA-2.1 electrode are presented before and after IR heating.  

- Many studies have reported the application of such composites to capacitors. Discuss the numerical comparison of the superiority of the capacitors prepared by this method compared to previous reports. 

Hybrid composites based on conjugated polymers and reduced graphene oxide have been studied exclusively as cathode materials for supercapacitors in aqueous sulfuric acid and alkaline electrolytes. We have not been able to find any references to the study of such composites in organic electrolytes with lithium salts. The present work is the first study of the electrochemical behavior of a cathode material based on a conductive polymer and reduced graphene oxide in a lithium organic electrolyte. Such cathode materials in organic electrolytes are the most promising for the creation of hybrid supercapacitors due to the possibility of increasing the supercapacitor voltage and achieving high values of energy density and charge-discharge currents.

The combination of a porous carbon substrate with high surface and new electroactive composite coatings in electrode materials makes it possible to balance contributions of double layer charging and Faraday pseudocapacitance in order to obtain both high capacitance and high charge-discharge currents. Poly-N-phenylanthranilic acid, synthesized by the authors for the first time, is well adsorbed on graphene nanosheets due to p-stacking and hydrogen bonding of carboxylic groups [55, 56] to oxygen-containing groups on the graphene oxide. With further recovery of graphene oxide and removal of oxygen, the polymer layers prevent aggregation of graphene nanosheets.

- Unless these are corrected, this paper should not be accepted for publication in this journal.

Appropriate additions and corrections are introduced into the text in colored characters.

Reviewer 3 Report

In this manuscript, the authors investigated the RGO/P-N-PAA composites for electroactive coatings were prepared by two different ways. However, the characteristic of materials is insufficient. There are so many mistakes in grammar or in format. The authors should carefully read the guide for authors and carefully check this paper. Therefore, I think the novelty and performance of this work are limited, and hence I would recommend rejection of this manuscript.

Author Response

- In this manuscript, the authors investigated the RGO/P-N-PAA composites for electroactive coatings were prepared by two different ways. However, the characteristic of materials is insufficient. There are so many mistakes in grammar or in format. The authors should carefully read the guide for authors and carefully check this paper. Therefore, I think the novelty and performance of this work are limited, and hence I would recommend rejection of this manuscript.

Hybrid composites based on conjugated polymers and reduced graphene oxide have been studied exclusively as cathode materials for supercapacitors in aqueous sulfuric acid and alkaline electrolytes. We have not been able to find any references to the study of such composites in organic electrolytes with lithium salts. However, such cathode materials in organic electrolytes are the most promising for the creation of hybrid supercapacitors, in which an anode of lithium battery is used. As a result, it is possible to increase the voltage of the supercapacitor and achieve the energy density characteristic of lithium-ion batteries. Moreover, the main advantage over batteries are high charge-discharge currents. 

The combination of a porous carbon substrate with high surface and new electroactive composite coatings in electrode materials makes it possible to balance contributions of double layer charging and Faraday pseudocapacitance in order to obtain both high capacitance and high charge-discharge currents. Poly-N-phenylanthranilic acid, synthesized by the authors for the first time, is well adsorbed on graphene nanosheets due to p-stacking and hydrogen bonding of carboxylic groups [55, 56] to oxygen-containing groups on the graphene oxide. With further recovery of graphene oxide and removal of oxygen, the polymer layers prevent aggregation of graphene nanosheets. The use of hybrid composites as cathode electrode materials in power sources with organic electrolytes will increase the voltage on the power source.

We carefully checked this paper. Appropriate changes were made.

A professional translator has corrected typos and mistakes. Proofreading of the manuscript was carried out.

Reviewer 4 Report

The results presented in the submitted manuscript are interesting, but they have to be placed in the context of similar polymer-coated rGO-based capacitors to show whether the P-N-PAA is superior compared to other materials used for the same purpose. It should be explicitly said what is the reason for using phenylanthranilic acid and not, for example polypyrrole (https://doi.org/10.1016/j.ijcas.2013.09.001). Other comments:

- The soundness of the manuscript can be increased by addition of scheme of the synthesis of the investigated composites.

- Experiments (both characterisation and electrochemical investigation) should be amended with “blank” results, that is, P-N-PAA without rGO to show how the properties changed with addition of GO. Furthermore, XPS C1s scans for IR-treated composites should be shown to see the rate of GO reduction.

- FTIR spectra should be grouped into one graph for each composite before and after IR treatment to make the change in the spectrum caused by the reduction more visible.

- What is the capacitance of composite with P-N-PAA2.2 before IR reduction? (It is not listed in the Table 4)

- What is the weight of the modifying composited used for electrodes displayed in figures 8 and 9?

- If Cw of AGF/RGO/P-N-PAA-1IR (0.38 mg) is 216 F g-1 and Cw of AGF/RGO/P-N-PAA-2.1IR (0.35 mg) is higher (377 F g-1), isn´t it reasonable to assume that after decreasing the amount of the composite (e.g., to 0.11 mg as in the case of AGF/RGO/P-N-PAA-1IR) the AGF/RGO/P-N-PAA-2.1IR would outperform AGF/RGO/P-N-PAA-1IR?

Author Response

- The results presented in the submitted manuscript are interesting, but they have to be placed in the context of similar polymer-coated rGO-based capacitors to show whether the P-N-PAA is superior compared to other materials used for the same purpose. It should be explicitly said what is the reason for using phenylanthranilic acid and not, for example polypyrrole (https://doi.org/10.1016/j.ijcas.2013.09.001).

Hybrid composites based on conjugated polymers and reduced graphene oxide have been studied exclusively as cathode materials for supercapacitors in aqueous sulfuric acid and alkaline electrolytes. The proposed reference presents the results of the study of the electrochemical behavior of the nanocomposite based on polypyrrole and reduced graphene oxide in an aqueous alkaline (KOH) electrolyte. We have not been able to find any references to the study of such composites in organic electrolytes with lithium salts. The present work is the first study of the electrochemical behavior of a cathode material based on a conductive polymer and reduced graphene oxide in a lithium organic electrolyte. Such cathode materials in organic electrolytes are the most promising for the creation of hybrid supercapacitors due to the possibility of increasing the supercapacitor voltage and achieving high values of energy density and charge-discharge currents.

The combination of a porous carbon substrate with high surface and new electroactive composite coatings in electrode materials makes it possible to balance contributions of double layer charging and Faraday pseudocapacitance in order to obtain both high capacitance and high charge-discharge currents. Poly-N-phenylanthranilic acid, synthesized by the authors for the first time, is well adsorbed on graphene nanosheets due to p-stacking and hydrogen bonding of carboxylic groups [55, 56] to oxygen-containing groups on the graphene oxide. With further recovery of graphene oxide and removal of oxygen, the polymer layers prevent aggregation of graphene nanosheets.

Proposed reference [41] was cited. Appropriate additions were introduced into the text in colored characters.

- The soundness of the manuscript can be increased by addition of scheme of the synthesis of the investigated composites.

Scheme of the RGO/P-N-PAA materials synthesis was added.

- Experiments (both characterisation and electrochemical investigation) should be amended with “blank” results, that is, P-N-PAA without rGO to show how the properties changed with addition of GO. Furthermore, XPS C1s scans for IR-treated composites should be shown to see the rate of GO reduction.

Appropriate additions were introduced into the text in colored characters.

- FTIR spectra should be grouped into one graph for each composite before and after IR treatment to make the change in the spectrum caused by the reduction more visible.

Appropriate changes were made.

- What is the capacitance of composite with P-N-PAA2.2 before IR reduction? (It is not listed in the Table 4).

The RGO(20%)/P-N-PAA-2.1 and RGO(50%)/P-N-PAA-2.2 composites were obtained by the same method and differ only in RGO content. When comparing the CV and capacitance data obtained from the charge-discharge curves of these composites after IR heating, it turned out that introducing more RGO decreases the capacitance characteristics of the electrode. Therefore, the AGF/RGO(50%)/P-N-PAA-2.2 electrode was not investigated, assuming that the capacitance of the initial coating would be lowered compared to RGO(20%)/P-N-PAA-2.1 because of the reduced Faraday capacitance contribution from the polymer, since its content in this composite was reduced by 30%.

The data for the AGF/RGO(20%)/P-N-PAA-2.1 electrode are presented before and after IR heating.

- What is the weight of the modifying composited used for electrodes displayed in figures 8 and 9?

Figures 8 and 9 present the AGF/RGO/P-N-PAA electrodes at commensurate weights of polymer and composite coatings to AGF. The corresponding weights are shown in the figures.

- If Cw of AGF/RGO/P-N-PAA-1IR (0.38 mg) is 216 F g-1 and Cw of AGF/RGO/P-N-PAA-2.1IR (0.35 mg) is higher (377 F g-1), isn´t it reasonable to assume that after decreasing the amount of the composite (e.g., to 0.11 mg as in the case of AGF/RGO/P-N-PAA-1IR) the AGF/RGO/P-N-PAA-2.1IR would outperform AGF/RGO/P-N-PAA-1IR?

We agree, because the capacitance is increased when the coatings are applied with a low weight. Comparison of both AGF/RGO/P-N-PAA-1, AGF/RGO/P-N-PAA-2.1 and AGF/RGO/P-N-PAA-1IR, AGF/RGO/P-N-PAA-2.1IR electrodes with comparable weights of 0.37, 0.29, 0.38 and 0.35 mg coatings supports our conclusions about higher capacitance characteristics for composite coating prepared from a solution of P-N-PAA polymer in DMF containing GO.

Round 2

Reviewer 2 Report

The papers are well-revised according to the reviewers' comments. Thus, the reviewer concluded that this paper may be acceptable for the publication in the journal "Polymers" in present form.

Author Response

The authors are grateful to the reviewer for constructive and valuable comments on the manuscript and decision to accept our article. 

Reviewer 3 Report

The authors should carefully  check  the mistakes in grammar.

Author Response

We have once again carefully checked the mistakes in grammar. Appropriate changes were made in color character.

Reviewer 4 Report

The revision has been performed according to most comments and questions, although few minor issues left:

Hybrid composites based on conjugated polymers and reduced graphene oxide have been studied exclusively as cathode materials for supercapacitors in aqueous sulfuric acid and alkaline electrolytes. The proposed reference presents the results of the study of the electrochemical behavior of the nanocomposite based on polypyrrole and reduced graphene oxide in an aqueous alkaline (KOH) electrolyte. We have not been able to find any references to the study of such composites in organic electrolytes with lithium salts. The present work is the first study of the electrochemical behavior of a cathode material based on a conductive polymer and reduced graphene oxide in a lithium organic electrolyte. Such cathode materials in organic electrolytes are the most promising for the creation of hybrid supercapacitors due to the possibility of increasing the supercapacitor voltage and achieving high values of energy density and charge-discharge currents.” – why is not this (basically the only one) explanation of novelty of the manuscript part of the text?

Fig. 2 caption – 1,3,5 is Ar and 2,4,6 is air, it should be corrected

“…When comparing the CV and capacitance data obtained from the charge-discharge curves of these composites after IR heating, it turned out that introducing more RGO decreases the capacitance characteristics of the electrode. Therefore, the AGF/RGO(50%)/P-N-PAA-2.2 electrode was not investigated, assuming that the capacitance of the initial coating would be lowered compared to RGO(20%)/P-N-PAA-2.1 because of the reduced Faraday capacitance contribution from the polymer, since its content in this composite was reduced by 30%.” This explanation should be part of the text.

We agree, because the capacitance is increased when the coatings are applied with a low weight. Comparison of both AGF/RGO/P-N-PAA-1, AGF/RGO/P-N-PAA-2.1 and AGF/RGO/P-N-PAA-1IR, AGF/RGO/P-N-PAA-2.1IR electrodes with comparable weights of 0.37, 0.29, 0.38 and 0.35 mg coatings supports our conclusions about higher capacitance characteristics for composite coating prepared from a solution of P-N-PAA polymer in DMF containing GO.” This response should be also part of the text.

Author Response

The authors are grateful to the reviewer for constructive and valuable comments on the manuscript. We fully agree with the reviewer.

Appropriate additions have been made to the text in colored characters.

Appropriate changes in Fig. 2 caption were made in color characters.